# UNSUPERVISED LEARNING OF GLOBAL FACTORS IN DEEP GENERATIVE MODELS

## ABSTRACT

We present a novel deep generative model based on non i.i.d. variational autoencoders that captures global dependencies among observations in a fully unsupervised fashion. In contrast to the recent semi-supervised alternatives for global modeling in deep generative models, our approach combines a mixture model in the local or data-dependent space and a global Gaussian latent variable, which lead us to obtain three particular insights. First, the induced latent global space captures interpretable disentangled representations with no user-defined regularization in the evidence lower bound (as in beta-VAE and its generalizations). Second, we show that the model performs domain alignment to find correlations and interpolate between different databases. Finally, we study the ability of the global space to discriminate between groups of observations with non-trivial underlying structures, such as face images with shared attributes or defined sequences of digits images.

## 1 INTRODUCTION

Since its first proposal by Kingma & Welling (2013), Variational Autoencoders (VAEs) have evolved into a vast amount of variants. To name some representative examples, we can include VAEs with latent mixture models priors (Dilokthanakul et al. (2016)), adapted to model time-series (Chung et al. (2015)), trained via deep hierarchical variational families (Ranganath et al. (2016), Tomczak & Welling (2018)), or that naturally handle heterogeneous data types and missing data (Nazabal et al. (2020)).

The large majority of VAE-like models are designed over the assumption that data is i.i.d., which remains a valid strategy for simplifying the learning and inference processes in generative models with latent variables. A different modelling approach may drop the i.i.d. assumption with the goal of capturing a higher level of dependence between samples. Inferring such kind of higher level dependencies can directly improve current approaches to find interpretable disentangled generative models (Bouchacourt et al. (2018)), to perform domain alignment (Heinze-Deml & Meinshausen (2017)) or to ensure fairness and unbiased data (Barocas et al. (2017)).

The main contribution of this paper is to show that a deep probabilistic VAE non i.i.d. model with both local and global latent variable can capture meaningful and interpretable correlation among data points in a completely unsupervised fashion. Namely, weak supervision to group the data samples is not required. In the following we refer to our model as Unsupervised Global VAE (UG-VAE). We combine a clustering inducing mixture model prior in the local space, that helps to separate the fundamental data features that an i.i.d. VAE would separate, with a global latent variable that modulates the properties of such latent clusters depending on the observed samples, capturing fundamental and interpretable data features. We demonstrate such a result using both CelebA, MNIST and the 3D FACES dataset in Paysan et al. (2009). Furthermore, we show that the global latent space can explain common features in samples coming from two different databases without requiring any domain label for each sample, establishing a probabilistic unsupervised framework for domain alignment. Up to our knowledge, UG-VAE is the first VAE model in the literature that performs unsupervised domain alignment using global latent variables.

Finally, we demonstrate that, even when the model parameters have been trained using an unsupervised approach, the global latent space in UG-VAE can discriminate groups of samples with non-trivial structures, separating groups of people with black and blond hair in CelebA or series of

numbers in MNIST. In other words, if weak supervision is applied at test time, the posterior distribution of the global latent variable provides with an informative representation of the user defined groups of correlated data.

## 2 RELATED WORK

Non i.i.d. deep generative models are getting recent attention but the literature is still scarse. First we find VAE models that implement non-parametric priors: in Gyawali et al. (2019) the authors make use of a global latent variable that induces a non-parametric Beta process prior, and more efficient variational mechanism for this kind of IBP prior are introduced in Xu et al. (2019). Second, both Tang et al. (2019) and Korshunova et al. (2018) proposed non i.i.d. exchangable models by including correlation information between datapoints via an undirected graph. Finally, some other works rely on simpler generative models (compared to these previous approaches), including global variables with fixed-complexity priors, typically a multi-variate Gaussian distribution, that aim at modelling the correlation between user-specified groups of correlated samples (e.g. images of the same class in MNIST, or faces of the same person). In Bouchacourt et al. (2018) or Hosoya (2019), authors apply weak supervision by grouping image samples by identity, and include in the probabilistic model a global latent variable for each of these groups, along with a local latent variable that models the distribution for each individual sample. Below we specify the two most relevant lines of research, in relation to our work.

**VAEs with mixture priors.** Several previous works have demonstrated that incorporating a mixture in the latent space leads to learn significantly better models. In Johnson et al. (2016) authors introduce a latent GMM prior with nonlinear observations, where the means are learned and remain invariant with the data. The GMVAE proposal by Dilokthanakul et al. (2016) aims at incorporating unsupervised clustering in deep generative models for increasing interpretability. In the VAMP VAE model Tomczak & Welling (2018), the authors define the prior as a mixture with components given by approximated variational posteriors, that are conditioned on learnable pseudo-inputs. This approach leads to an improved performance, avoiding typical local optima difficulties that might be related to irrelevant latent dimensions.

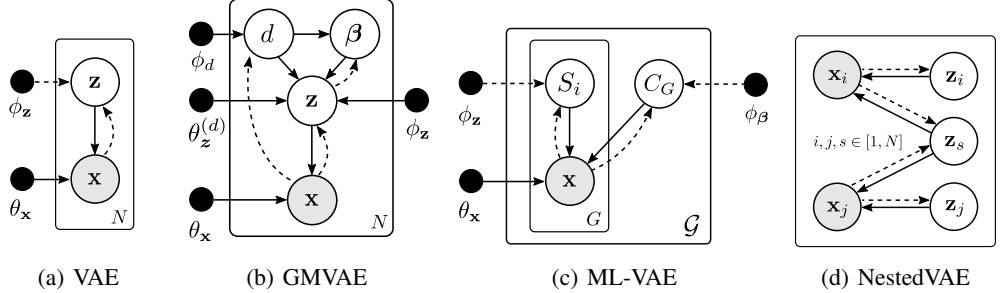

(a) VAE     (b) GMVAE     (c) ML-VAE     (d) NestedVAE

Figure 1: Comparison of four deep generative models. Dashed lines represent the graphical model of the associated variational family. The Vanilla VAE (a), the GMVAE (b), and semi-supervised variants for grouped data; ML-VAE (c) and NestedVAE (d).

**Semi-supervised deep models for grouped data.** In contrast to the i.i.d. vanilla VAE model in Figure 1 (a), and its augmented version for unsupervised clustering, GMVAE, in Figure 1 (b), the graphical model of the Multi-Level Variational Autoencoder (ML-VAE) in Bouchacourt et al. (2018) is shown in Figure 1 (c), where G denotes the number of groups. ML-VAE includes a local Gaussian variable $S_i$ that encodes style-related information for each sample, and global Gaussian variable $C_G$ to model shared in a group of samples. For instance, they feed their algorithm with batches of face images from the same person, modeling content shared within the group that characterize a person. This approach leads to learning a disentangled representations at the group and observations level, in a content-style fashion. Nevertheless, the groups are user-specified, hence resulting in a semi-supervised modelling approach. In Vowels et al. (2020) authors use weak supervision for pairing samples. They implement two outer VAEs with shared weights for the reconstruction, and a Nested

VAE that reconstructs latent representation off one to another, modelling correlations across pairs of samples. The graphical model for Nested VAE is depicted in Figure 1 (d).

# 3 UNSUPERVISED GLOBAL VAE

We present UG-VAE, a deep generative VAE framework for modeling non-i.i.d. data with global dependencies. It generalizes the ML-VAE graphical model in Figure 1 (c) to *i)* remove the group supervision, *ii)* include a clustering-inducing prior in the local space, and *iii)* propose a more structured variational family.

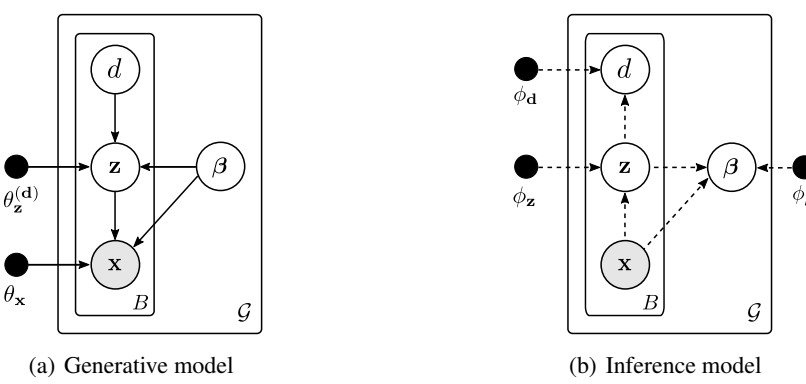

(a) Generative model                    (b) Inference model

Figure 2: Generative (left) and inference (right) of UG-VAE.

## 3.1 GENERATIVE MODEL

Figure 2 represents the generative graphical model of UG-VAE. A global variable $\boldsymbol{\beta} \in \mathbb{R}^g$ induces shared features to generate a group of $B$ samples $\mathbf{X} = \{\mathbf{x}_1, ..., \mathbf{x}_B\} \subseteq \mathbb{R}^D$, and $\mathcal{G}$ is the number of groups we jointly use to amortize the learning of the model parameters. During amortized variational training, groups are simply random data mini-batches from the training dataset, being $\mathcal{G}$ the number of data mini-batches. We could certainly take $B = N$ (the training set size) and hence $\mathcal{G} = 1$, but this leads to less interpretable global latent space (too much data to correlate with a single global random variable), and a slow training process.

Conditioned to $\boldsymbol{\beta}$, data samples are independent and distributed according to a Gaussian mixture local (one per data) latent variable $\mathbf{Z} = \{\mathbf{z}_1, ..., \mathbf{z}_B\} \subseteq \mathbb{R}^d$, and $\mathbf{d} = \{d_1, ..., d_B\} \subseteq \{1, ..., K\}$ are independent discrete categorical variables with uniform prior distributions. This prior, along with the conditional distribution $p(\mathbf{z}_i | d_i, \boldsymbol{\beta})$, defines a Gaussian mixture latent space, which helps to infer similarities between samples from different batches (by assigning them to the same cluster), and thus, $d_i$ plays a similar role than the semi-supervision included in Bouchacourt et al. (2018) by grouping. Our experimental results demonstrate that this level of structure in the local space is crucial to acquire interpretable information at the global space, and specially, if we fix $d_i$ for all the samples within a batch, that the global variable $\boldsymbol{\beta}$ is able to tune different generative factors for each cluster.

The joint distribution for a single group is therefore defined by:

$$p_\theta(\mathbf{X}, \mathbf{Z}, \mathbf{d}, \boldsymbol{\beta}) = p(\mathbf{X}|\mathbf{Z}, \boldsymbol{\beta}) \, p(\mathbf{Z}|\mathbf{d}, \boldsymbol{\beta}) \, p(\mathbf{d}) \, p(\boldsymbol{\beta}) \tag{1}$$

where the likelihood term of each sample is a Gaussian distribution, whose parameters are obtained from a concatenation of $\mathbf{z}_i$ and $\boldsymbol{\beta}$ as input of a decoder network:

$$p(\mathbf{X}|\mathbf{Z}, \boldsymbol{\beta}) = \prod_{i=1}^{B} p(\mathbf{x}_i|\mathbf{z}_i, \boldsymbol{\beta}) = \prod_{i=1}^{B} \mathcal{N}\left(\boldsymbol{\mu}_{\theta_x}([\mathbf{z}_i, \boldsymbol{\beta}]), \boldsymbol{\Sigma}_{\theta_x}([\mathbf{z}_i, \boldsymbol{\beta}])\right) \tag{2}$$

In contrast with Johnson et al. (2016), where the parameters of the clusters are learned but shared by all the observations, in UG-VAE, the parameters of each component are obtained with networks fed

with $\boldsymbol{\beta}$. Thus, the prior of each local latent continuous variable is defined by a mixture of Gaussians, where $d_i$ defines the component and $\boldsymbol{\beta}$ is the input of a NN that outputs its parameters:

$$p(\mathbf{Z}|\mathbf{d}, \boldsymbol{\beta}) = \prod_{i=1}^{B} p(\mathbf{z}_i|d_i, \boldsymbol{\beta}) = \prod_{i=1}^{B} \mathcal{N}\left(\boldsymbol{\mu}_{\theta_z}^{(d_i)}(\boldsymbol{\beta}), \boldsymbol{\Sigma}_{\theta_z}^{(d_i)}(\boldsymbol{\beta})\right), \tag{3}$$

hence we trained as many NNs as discrete categories. This local space encodes samples in representative clusters to model local factors of variation. The prior of the discrete latent variable is defined as uniform:

$$p(\mathbf{d}) = \prod_{i=1}^{B} \text{Cat}(\boldsymbol{\pi}) \quad \pi_k = 1/K \tag{4}$$

and the prior over the continuous latent variable $\beta$ follows an isotropic Gaussian, $p(\boldsymbol{\beta}) = \mathcal{N}(\mathbf{0}, \mathbf{I})$.

## 3.2 INFERENCE MODEL

The graphical model of the proposed variational family is shown in Figure 2(b):

$$q_\phi(\mathbf{Z}, \mathbf{d}, \boldsymbol{\beta}|\mathbf{X}) = q(\mathbf{Z}|\mathbf{X})\, q(\mathbf{d}|\mathbf{Z}) q(\boldsymbol{\beta}|\mathbf{X}, \mathbf{Z}) \tag{5}$$

where we employ an encoder network that maps the input data into the local latent posterior distribution, which is defined as a Gaussian:

$$q(\mathbf{Z}|\mathbf{X}) = \prod_{i=1}^{B} q(\mathbf{z}_i|\mathbf{x}_i) = \prod_{i=1}^{B} \mathcal{N}(\boldsymbol{\mu}_{\phi_z}(\mathbf{x}_i), \boldsymbol{\Sigma}_{\phi_z}(\mathbf{x}_i)) \tag{6}$$

Given the posterior distribution of $\mathbf{z}$, the categorical posterior distribution of $d_i$ is parametrized by a NN that takes $\mathbf{z}_i$ as input

$$q(\mathbf{d}|\mathbf{Z}) = \prod_{i=1}^{B} q(d_i|\mathbf{z}_i) = \prod_{i=1}^{B} \text{Cat}(\pi_{\phi_d}(\mathbf{z}_i)) \tag{7}$$

The approximate posterior distribution of the global variable $\boldsymbol{\beta}$ is computed as a product of local contributions per datapoint. This strategy, as demonstrated by Bouchacourt et al. (2018), outperforms other approaches like, for example, a mixture of local contributions, as it allows to accumulate group evidence. For each sample, a NN encodes $\mathbf{x}_i$ and the Categorical parameters $\pi_{\phi_d}(\mathbf{z}_i)$ in a local Gaussian:

$$q(\boldsymbol{\beta}|\mathbf{X}, \mathbf{Z}) = \mathcal{N}\left(\boldsymbol{\mu}_\beta, \boldsymbol{\Sigma}_\beta\right) = \prod_{i=1}^{B} \mathcal{N}\left(\boldsymbol{\mu}_{\phi_\beta}([\mathbf{x}_i, \pi_{\phi_d}(\mathbf{z}_i)]), \boldsymbol{\Sigma}_{\phi_\beta}([\mathbf{x}_i, \pi_{\phi_d}(\mathbf{z}_i)])\right) \tag{8}$$

If we denote by $\boldsymbol{\mu}_i$ and $\boldsymbol{\Sigma}_i$ the parameters obtained by networks $\boldsymbol{\mu}_{\phi_\beta}$ and $\boldsymbol{\Sigma}_{\phi_\beta}$, respectively, the parameters of the global Gaussian distribution are given, following Bromiley (2003), by:

$$\begin{aligned} \boldsymbol{\Lambda}_\beta &= \boldsymbol{\Sigma}_\beta^{-1} = \sum_{i=1}^{B} \boldsymbol{\Lambda}_i \\ \boldsymbol{\mu}_\beta &= (\boldsymbol{\Lambda}_\beta)^{-1} \sum_{i=1}^{B} \boldsymbol{\Lambda}_i \boldsymbol{\mu}_i \end{aligned} \tag{9}$$

where $\boldsymbol{\Lambda}_\beta = \boldsymbol{\Sigma}_\beta^{-1}$ is defined as the precision matrix, which we model as a diagonal matrix.

## 3.3 EVIDENCE LOWER BOUND

Overall, the evidence lower bound reads as follows:

$$\mathcal{L}(\theta, \phi; \mathbf{X}, \mathbf{Z}, \mathbf{d}, \boldsymbol{\beta}) = \mathbb{E}_{q(\boldsymbol{\beta})}\left[\mathcal{L}_i(\theta, \phi; \mathbf{x}_i, \mathbf{z}_i, \mathbf{d}, \boldsymbol{\beta})\right] - \mathbb{E}_{q(\mathbf{d})}\left[D_{KL}\left(q(\boldsymbol{\beta}|\mathbf{X}, \mathbf{Z})\|p(\boldsymbol{\beta})\right)\right] \tag{10}$$

The resulting ELBO is an expansion of the ELBO for a standard GMVAE with a new regularizer for the global variable. As the reader may appreciate, the ELBO for UG-VAE does not include extra

hyperparameters to enforce disentanglement, like other previous works as $\beta$-VAE, and thus, no extra validation is needed apart from the parameters of the networks architecture, the number of clusters and the latent dimensions. We denote by $\mathcal{L}_i$ each local contribution to the ELBO:

$$
\mathcal{L}_i(\theta, \phi; \mathbf{x}_i, \mathbf{z}_i, \mathbf{d}, \boldsymbol{\beta}) = \mathbb{E}_{q(\boldsymbol{d_i}, \boldsymbol{z}_i)} \left[ \log p(\mathbf{x}_i | \mathbf{z}_i, d_i, \boldsymbol{\beta}) \right]
$$
$$
- \mathbb{E}_{q(\boldsymbol{d_i})} \left[ D_{KL} \left( q(\mathbf{z}_i | \mathbf{x}_i) \| p(\mathbf{z}_i | d_i, \boldsymbol{\beta}) \right) \right] - D_{KL} \left( q(d_i | \mathbf{z}_i) \| p(d_i) \right)) \tag{11}
$$

The first part of equation 10 is an expectation over the global approximate posterior of the so-called local ELBO. This local ELBO differs from the vanilla ELBO proposed by Kingma & Welling (2013) in the regularizer for the discrete variable $d_i$, which is composed by the typical reconstruction term of each sample and two KL regularizers: one for $\mathbf{z}_i$, expected over $d_i$, and the other over $d_i$. The second part in equation 10 is a regularizer on the global posterior. The expectations over the discrete variable $d_i$ are tractable and thus, analytically marginalized.

In contrast with GMVAE (Figure 1 (b)), in UG-VAE, $\boldsymbol{\beta}$ is shared by a group of observations, therefore the parameters of the mixture are the same for all the samples in a batch. In this manner, within each optimization step, the encoder $q(\boldsymbol{\beta}|\mathbf{X}, \mathbf{Z})$ only learns from the global information obtained from the product of Gaussian contributions of every observation, with the aim at configuring the mixture to improve the representation of each datapoint in the batch, by means of $p(\mathbf{Z}|\boldsymbol{\beta}, \mathbf{X})$ and $p(\mathbf{X}|\mathbf{Z}, \boldsymbol{\beta})$. Hence, the control of the mixture is performed by using global information. In contrast with ML-VAE (whose encoder $q(C_G|\mathbf{X})$ is also global, but the model does not include a mixture), in UG-VAE, the $\boldsymbol{\beta}$ encoder incorporates information about which component each observation belongs to, as the weights of the mixture inferred by $q(d|\mathbf{Z})$ are used to obtain $q(\boldsymbol{\beta}|\mathbf{X}, \mathbf{Z})$. Thus, while each cluster will represent different local features, moving $\boldsymbol{\beta}$ will affect all the clusters. In other words, modifying $\boldsymbol{\beta}$ will have some effect in each local cluster. As the training progresses, the encoder $q(\boldsymbol{\beta}|\mathbf{X}, \mathbf{Z})$ learns which information emerging from each batch of data allows to move the cluster in a way that the ELBO increases.

## 4 EXPERIMENTS

In this section we demonstrate the ability of the UG-VAE model to infer global factors of variation that are common among samples, even when coming from different datasets. In all cases, we have not validated in depth all the networks used, we have merely rely on encoder/decoder networks proposed in state-of-the-art VAE papers such as Kingma & Welling (2013), Bouchacourt et al. (2018) or Higgins et al. (2016). Our results must be hence regarded as a proof of concept about the flexibility and representation power of UG-VAE, rather than fine-tuned results for each case. Hence there is room for improvement in all cases. Details about network architecture and training parameters are provided in the supplementary material.

### 4.1 UNSUPERVISED LEARNING OF GLOBAL FACTORS

In this section we first asses the interpretability of the global disentanglement features inferred by UG-VAE over both CelebA and MNIST. In Figure 3 we show samples of the generative model as we explore both the global and local latent spaces. We perform a linear interpolation with the aim at exploring the hypersphere centered at the mean of the distribution and with radius $\sigma_i$ for each dimension $i$. To maximize the variation range across every dimension, we move diagonally through the latent space. Rows correspond to an interpolation on the global $\boldsymbol{\beta}$ between $[-1, 1]$ on every dimension ($p(\boldsymbol{\beta})$ follows a standard Gaussian). As the local $p(\mathbf{z}|d, \boldsymbol{\beta})$ (equation 3) depends on $d$ and $\boldsymbol{\beta}$, if we denote $\boldsymbol{\mu}_z = \boldsymbol{\mu}_z^{(d)}(\boldsymbol{\beta})$, the local interpolation goes from $[\mu_{z0} - 3, \mu_{z1} - 3, ... \mu_{zd} - 3]$ to $[\mu_{z0} + 3, \mu_{z1} + 3, ..., \mu_{zd} + 3]$. The range of $\pm 3$ for the local interpolation is determined to cover the variances $\boldsymbol{\Sigma}_z^{(d)}(\boldsymbol{\beta})$ that we observe upon training the model for MNIST and CelebA. The every image in Figure 3 correspond to samples from a different cluster (fixed values of $d$), in order to facilitate the interpretability of the information captured at both local and global levels. By using this set up, we demonstrate that the global information tuned by $\boldsymbol{\beta}$ is different and clearly interpretable inside each cluster.

The total number of clusters is set to $K = 20$ for CelebA and $K = 10$ for MNIST. Three of these components are presented in Figure 3. We can observe that each row (each value of $\boldsymbol{\beta}$) induces a

shared generative factor, while **z** is in charge of variations inside this common feature. For instance, in CelebA (top), features like skin color, presence of beard or face contrast are encoded by the global variable, while local variations like hair style or light direction are controlled by the local variable. In a simple dataset like MNIST (bottom), results show that handwriting global features as cursive style, contrast or thickness are encoded by $\beta$, while the local **z** defines the shape of the digit. The characterization of whether these generative factors are local/global is based on an interpretation of the effect that varying **z** and $\beta$ provokes in each image within a batch, and in the whole batch of images, respectively. In the supplementary material, we reproduce the same figures for the all the clusters, in which we can appreciate that there is a significant fraction of clusters with visually interpretable global/local features.

We stress here again the fact that the UG-VAE training is fully unsupervised: data batches during training are completely randomly chosen from the training dataset, with no structured correlation whatsoever. Unlike other approaches for disentanglement, see Higgins et al. (2016) or Mathieu et al. (2019), variational training in UG-VAE does not come with additional ELBO hyperparameters that need to be tuned to find a proper balance among terms in the ELBO.

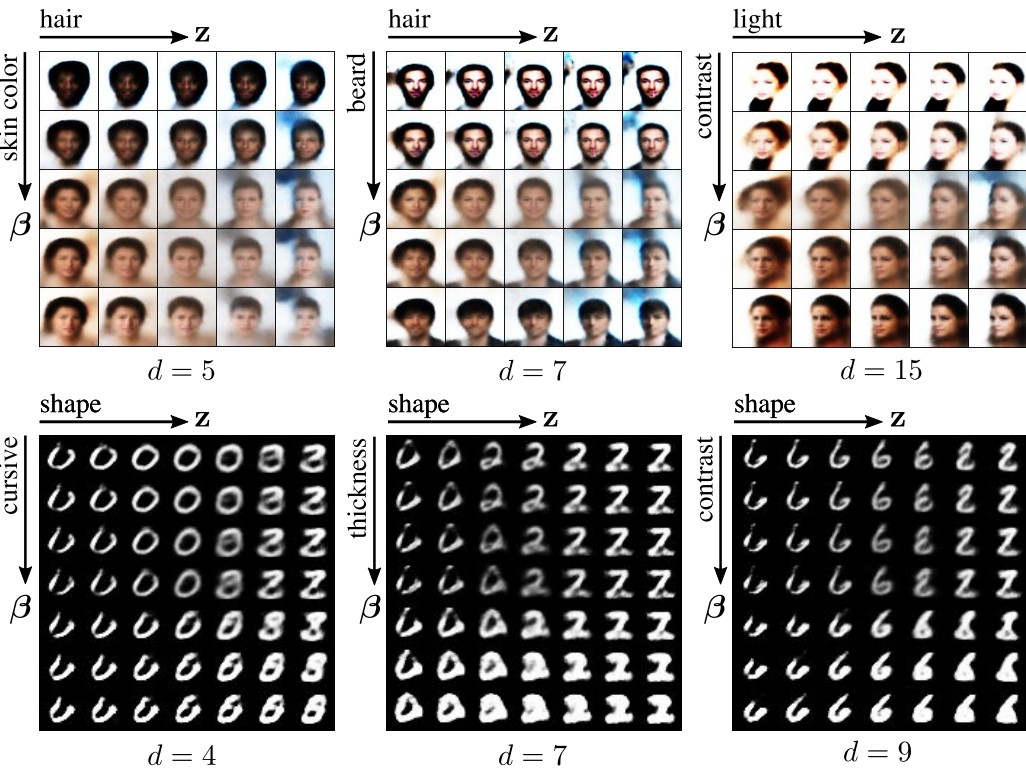

Figure 3: Sampling from UG-VAE for CelebA (top) and MNIST (bottom). We include samples from 3 local clusters from a total of $K = 20$ for CelebA and $K = 10$ for MNIST. In CelebA (top), the global latent variable disentangles in skin color, beard and face contrast, while the local latent variable controls hair and light orientation. In MNIST (bottom), $\beta$ controls cursive grade, contrast and thickness of handwriting, while **z** varies digit shape.

One of the main contributions in the design of UG-VAE is the fact that, unless we include a clustering mixture prior in the local space controlled by the global variable $\beta$, unsupervised learning of global factors is non-informative. To illustrate such a result, in Figure 4 we reproduce the results in Figure 3 but for a probabilistic model in which the discrete local variable $d$ is not included. Namely, we use the ML-VAE in Figure 2(c) but we trained it with random data batches. In this case, the local space is uni-modal given $\beta$ and we show interpolated values between -1 to 1. Note that the disentanglement effect of variations in both $\beta$ and $z$ is mild and hard to interpret.

z

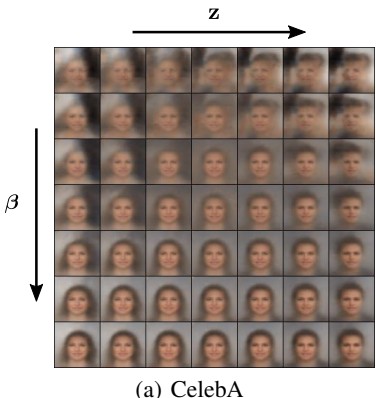
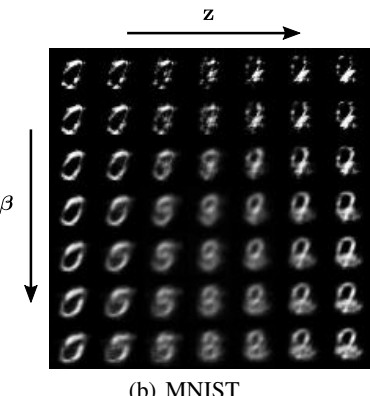

β                                    β

(a) CelebA                          (b) MNIST

Figure 4: Sampling from ML-VAE, trained over unsupervised data.

## 4.2 DOMAIN ALIGNMENT

In this section, we evaluate the UG-VAE performance in an unsupervised domain alignment setup. During training, the model is fed with data batches that include random samples coming from two different datasets. In particular, we train our model with a mixed dataset between CelebA and 3D FACES Paysan et al. (2009), a dataset of 3D scanned faces, with a proportion of 50% samples from each dataset inside each batch.

Upon training with random batches, in Figure 5, we perform the following experiment using domain supervision to create test data batches. We create two batches containing only images from CelebA and 3D FACES. Let $\beta_1$ and $\beta_2$ be the mean global posterior computed using (8) associated for each batch. For two particular images in these two batches, let $z_1$ and $z_2$ be the mean local posterior of these two images, computed using (3). Figure 5 (a) shows samples of the UG-VAE model when we linearly interpolate between $\beta_1$ and $\beta_2$ (rows) and between $z_1$ and $z_2$ (columns)[1]. Certainly $\beta$ is capturing the domain knowledge. For fixed z, e.g. $z_1$ in the first column, the interpolation between $\beta_1$ and $\beta_2$ is transferring the CelebA image into the 3D FACES domain (note that background is turning white, and the image is rotated to get a 3D effect). Alternatively, for fixed $\beta$, e.g. $\beta_1$ in the first row, interpolating between $z_1$ and $z_2$ modifies the first image into one that keeps the domain but resembles features of the image in the second domain, as face rotation.

In Figure 5(b) we show the 2D t-SNE plot of the posterior distribution of $\beta$ for batches that are random mixtures between datasets (grey points), batches that contain only CelebA faces (blue squares), and batches that contain only 3D faces (green triangles). We also add the corresponding points of the $\beta_1$ and $\beta_2$ interpolation in Figure 5(a). In Figure 5(c), we reproduce the experiment in (a) but interpolating between two images and values of $\beta$ that correspond to the same domain (brown interpolation line in Figure 5(b)). As expected, the interpolation of $\beta$ in this case does not change the domain, which suggests that the domain structure in the global space is smooth, and that the interpolation along the local space z modifies image features to translate one image into the other. In Figure 6 experiments with more datasets are included. When mixing the 3DCars dataset (Fidler et al. (2012)) with the 3D Chairs dataset (Aubry et al. (2014)), we find that certain correlations between cars and chairs are captured. In Figure 6 (a), interpolating between a racing car and an office desk chair leads to a white car in the first domain (top right) and in a couch (bottom left). In Figure 6 (b), when using the 3D Cars along with the Cars Dataset (Krause et al. (2013)), rotations in the cars are induced.

Finally, in the supplementary material we show that, as expected, the rich structured captured by UG-VAE illustrated in Figure 5 is lost when we do not include the clustering effect in the local space, i.e. if we use ML-VAE with unsupervised random data batches, and all the transition between domains is performed within the local space.

---

[1]Note that since both $\beta$ and $z$ are deterministically interpolated, the discrete variable $d$ plays no role to sample from the model.

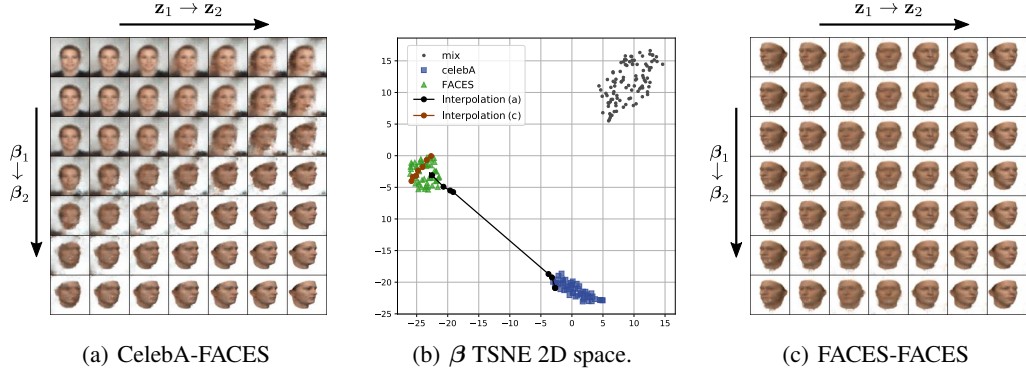

(a) CelebA-FACES     (b) $\boldsymbol{\beta}$ TSNE 2D space.     (c) FACES-FACES

Figure 5: UG-VAE interpolation in local (columns) and global (rows) posterior spaces, fusing celebA and FACES datasets. In (a) the interpolation goes between the posteriors of a sample from CelebA dataset and a sample from FACES dataset. In (c) the interpolation goes between the posteriors of a sample from FACES dataset and another sample from the same dataset.

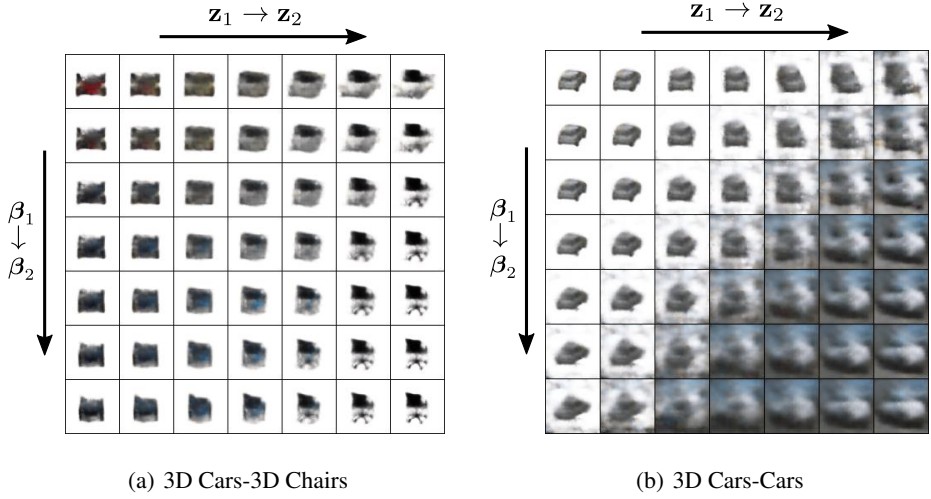

(a) 3D Cars-3D Chairs         (b) 3D Cars-Cars

Figure 6: Extended experiment: UG-VAE interpolation in local (columns) and global (rows) posterior spaces, fusing 3D Cars with 3D Chairs (d) and 3D Cars to Cars Dataset (e).

## 4.3 UG-VAE REPRESENTATION OF STRUCTURED NON-TRIVIAL DATA BATCHES

In the previous subsection, we showed that the UG-VAE global space is able to separate certain structure in the data batches (e.g. data domain) even though during training batches did not present such an explicit correlation. Using UG-VAE trained over CelebA with unsupervised random batches of 128 images as a running example, in this section we want to further demonstrate this result.

In Figure 7 we show the t-SNE 2D projection of structured batches using the posterior $\boldsymbol{\beta}$ distribution in (8) over CelebA test images. In Figure 7(a), we display the distribution of batches containing only men and women, while in Figure 7(b) the distribution of batches containing people with black or blond hair. In both cases we show the distribution of randomly constructed batches as the ones in the training set. To some extend, in both cases we obtain separable distributions among the different kinds of batches. A quantitive evaluation can be found in Table 1, in which we have used samples from the $\boldsymbol{\beta}$ distribution to train a supervised classifier to differentiate between different types of batches. When random batches are not taken as a class, the separability is evident. When random batches are included, it is expected that the classifier struggles to differentiate between a batch that contains 90% of male images and a batch that only contain male images, hence the drop in accuracy for the multi-case problem.

An extension with similar results and figures for another interpretation of global information capturing are exposed in the supplementary material, using structured grouped batches in MNIST dataset. In this experiment, the groups are digits that belong to certain mathematical series, including even numbers, odd numbers, Fibonacci series and prime numbers, and we prove that the model is able to discriminate among their global posterior representations.

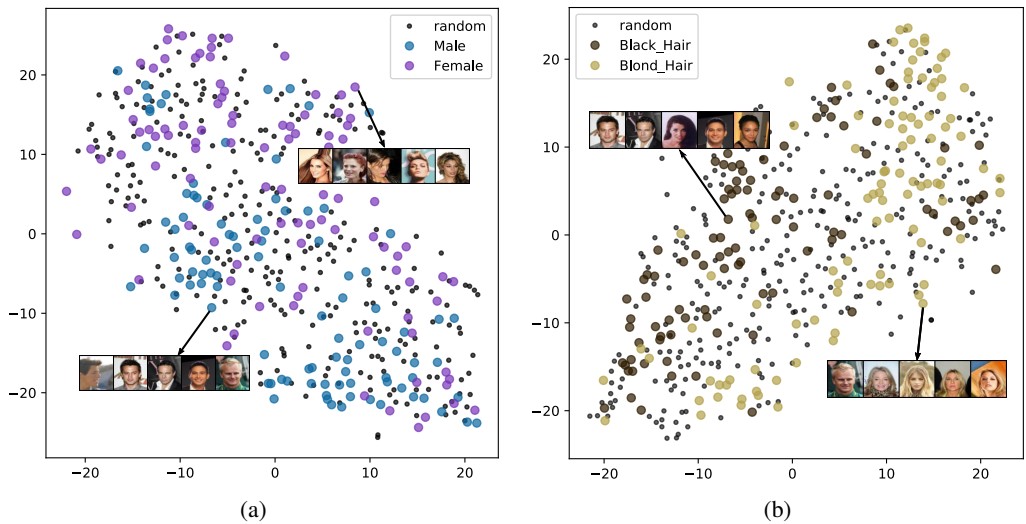

(a)               (b)

Figure 7: 2D t-SNE projection of the UG-VAE $\beta$ posterior distribution of structured batches of 128 CelebA images. UG-VAE is trained with completely random batches of 128 train images.

Table 1: Batch classification accuracy using samples of the posterior $\beta$ distribution.

| Batch categories | Classifier | Train accuracy | Test accuracy |
|---|---|---|---|
| Black (0) vs blond (1) | Linear SVM | 1.0 | 0.95 |
| | RBF SVM | 1.0 | 0.98 |
| Black (0) vs blond (1) vs random (2) | Linear SVM | 0.91 | 0.54 |
| | RBF SVM | 0.85 | 0.56 |
| Male (0) vs female (1) | Linear SVM | 1.0 | 0.85 |
| | RBF SVM | 1.0 | 0.85 |
| Male (0) vs female (1) vs random (2) | Linear SVM | 0.84 | 0.66 |
| | RBF SVM | 0.89 | 0.63 |

## 5  CONCLUSION

In this paper we have presented UG-VAE, an unsupervised generative probabilistic model able to capture both local data features and global features among batches of data samples. Unlike similar approaches in the literature, by combining a structured clustering prior in the local latent space with a global latent space with Gaussian prior and a more structured variational family, we have demonstrated that interpretable group features can be inferred from the global latent space in a completely unsupervised fashion. Model training does not require artificial manipulation of the ELBO term to force latent interpretability, which makes UG-VAE stand out w.r.t. most of the current disentanglement approaches using VAEs. The ability of UG-VAE to infer diverse features from the training set is further demonstrated in a domain alignment setup, where we show that the global space allows interpolation between domains, and also by showing that images in correlated batches of data, related by non-trivial features such as hair color or gender in CelebA, define identifiable structures in the posterior global latent space distribution.

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
