# OpenReview forum: "Unsupervised Learning of Global Factors in Deep Generative Models"
_ICLR.cc/2021/Conference — Reject_

### Official Review · AnonReviewer2 · 2020-10-26
**A solid paper**

**Rating:** 6
**Confidence:** 3

**Review:**

This paper proposed a deep generative model based on the non i.i.d. VAE framework in an unsupervised version. The model which combines a mixture prior in the local latent space with global latent space has three advantages: First, the latent space can capture interpretable features. Second, the model performs domain alignment. Third, the model can discriminate among their global posterior representations. Although this paper has mild improvement on the basic VAE structure, the model displays a good interpretability power, and the setup of the latent variables are illustrated reasonably in the paper.

Strength:
1. This paper models on non-i.i.d data in an unsupervised version, which provides a flexible model.
2. This model provides a good interpretability power. This paper demonstrates how the features are controlled by the global and local variables, and also it shows the necessary of including the mixture prior in the local space to acquire interpretable information.
3. This model performs domain alignment, the global variable β can capture domain knowledge.

Critiques:
1. The paper only discussed one prior distribution for the latent variable d and β. The power of choosing other types of prior distribution is unknown.
2. For the domain alignment experiment, only two datasets are used, it will be more convincing to include more datasets.

I have read authors' feedback and will keep my original score.

---

> ### Author Response · Authors · 2020-11-17
> **Response to Reviewer 2**
>
> We would like to thank the reviewer for the positive feedback and thoughtful comments. In the following, we carefully address all the critiques.
>
> 1. Certainly, a more flexible choice for these priors would drive into more powerful/interpretable models. Our following work will probably be centered on the discussion mentioned by the reviewer about how choosing different priors might improve results provided. Nevertheless, as the idea of the paper is to introduce our model and remark the novelty, we believe it is out of the scope of the paper at this point. Thank you for your suggestion.
>
> 2. Following your comment and a similar one by another reviewer, we have expanded these results with another two new examples in section 4.2.

---

### Official Review · AnonReviewer3 · 2020-10-27
**Interesting article, but the theoretical justification needs to be clarified**

**Rating:** 5
**Confidence:** 4

**Review:**

This article introduces a VAE-based method for separating local variation factors from global variation factors in the data in an unsupervised manner. It achieves so by designing a graphical model with a mix of example-local and batch-shared variables, and training it using the ELBO. The article provide an detailed experimental analysis on MNIST and CelebA, and experimental evidence that all parts of the model (notably the discrete d variable) are relevant.

The article provides a well detailed description of the proposed UG-VAE, and how it compare to similar models from the literature (notably ML-VAE, from which it is inspired). The experimental analysis is reasonably convincing, though the interpretation of the provided figures in the text is a little more optimistic than I would agree.

There is however one point in particular I would like to see clarified for this paper to be accepted: the justification of the structural design. I have several questions/remarks regarding it:

1. Equation 8 shows that q(β | X, d) actually depends on the categorical parameters of q(d|Z), rather than the value of d itself. As such, from a correctness perspective, the distribution is q(β | X, Z) (with weight sharing with q(d | Z)): it does not depend on the value of d. I suspect this choice was made to escape the computational cost of marginalizing the whole batch of variables d in the joint distribution for β, but this changes the meaning of the model.

2. The analysis of what features get stored in β versus Z is not very insightful. The text of the paper present this as if it was obvious, but does not seem that obvious to me why "beard" is a global feature but "hair" is a local one. The same non-obviousness applies to all examples.

3. I don't get why the training procedure is supposed to work. It empirically does to some extent, but there is no clear theoretical justification. If the same β is used for the whole batch, given the batch is randomly sampled, what part of the dynamic would indeed drive the model toward extracting some "common" features into this variable?

For this last point in particular, the proposed structure seems pretty close to an other one that would be I believe much more natural: have the global parameter β not shared across the entire batch, but depending on the class d the encoder assigned to the datapoint. Is that something that has been considered, and if yes why was it not satisfactory compared to the proposed model?

I believe it is necessary to clearly address these points for the article to be accepted.

--------

Small remarks & typos:

- Just after equation 9, there is a ^-1 missing on the sigma in the text
- the notation for the KL-divergence is inconsistent between equations 10 and 11 ( D_KL(...) vs KL(...) )
- the text in 4.A about how the z feature is explored in figure 3 is not completely clear. Is the interpolation done only along the diagonal, moving from (μ1 - 3, μ2 - 3, ...) to (μ1 + 3, μ2 + 3, ...) ? If so, why ? If it is rather only moved along one particular dimension, then I suggest rewording this sentence to make it clear.

---

> ### Author Response · Authors · 2020-11-17
> **Response to Reviewer 3: PART 1**
>
> We would like to thank the reviewer for the valuable feedback and comments. Regarding the justification of the structural design:
>
> 1. The reviewer is right, the notation might result confusing. We have replaced $q(\boldsymbol{\beta} | \mathbf{X}, \mathbf{d})$ by $q(\boldsymbol{\beta} | \mathbf{X}, \mathbf{Z})$, and the diagram of the inference model has also been corrected. Thanks for pointing out this issue.
>
> 2. The concept of local and global, in our work, lies on the fact that the information is extracted from a single image or from a group of images, independently on the semantics of the generative factor. We understand that in other works (like [1], [2], [3]), these concepts are based on the nature of the features, and the global-local features are referred, for example, as style-content. In our experiment, we use the generative model for sampling a batch of images (fixing $d$ to help the interpretations). The reason why we call ‘light’ or ‘hair’ a local factor is that we interpret this disentanglement after varying the local latent variable and fixing the global $\boldsymbol{\beta}$ for every image within a batch, for $d=15$ and $d=7$, respectively. On the other hand, ‘contrast’ is defined as a global factor after interpreting how every image within a batch changes when varying $\boldsymbol{\beta}$ with $\mathbf{z}$ fixed. In the second plot top row in Figure 3, we selected the 7th cluster (out of 20) and from samples we can interpret that for $\mathbf{z}$ fixed, the variation of $\boldsymbol{\beta}$ happens to control the presence of beard, hence we assume it is a global factor, and for $\boldsymbol{\beta}$ fixed, the variation of $\mathbf{z}$ controls the amount of hair the person has. This is an interpretation that we draw from samples of the model as we move around each of the two latent spaces.
>  We have extended the second paragraph of section 4.1 with this explanation.
>
> 3. The global latent variable $\boldsymbol{\beta}$ controls the cluster means and variances of the Gaussian mixture prior we use to sample every image. As such, it provides additional flexibility to group data points in the same minibatch. Furthermore, the generation of each image depends on both $\mathbf{z}$ and $\boldsymbol{\beta}$, and this enables a direct control of the image features according to the global feature. When we mix two types of datapoints (e.g. CelebA and FACES in Section 4.2), for a certain region of the $\boldsymbol{\beta}$ space, the clusters are merely separating CelebA faces, while for other $\boldsymbol{\beta}$ values they might be clustering according to correlations between images in CelebA and images in FACES.
>  On the other hand, the posterior distribution of $\boldsymbol{\beta}$ in equation (8) is an additive contribution of all the local embedding $\mathbf{z}$ values in the mini-batch. We believe this is crucial to extract common features into the $\boldsymbol{\beta}$ variable.
>  In comparison with ML-VAE (where they feed the model with grouped data), the hidden variable $d$ allows us to infer the group that each sample might belong to when the data is feeded randomly. The $d$ can find correlations between samples from different batches by assigning them to the same cluster. If the $d$ is removed, we obtain the ML-VAE, and this type of global information can only be captured in the case we include semi-supervision (Figure 4 corroborates this, as the disentanglement effect of variations in both $\boldsymbol{\beta}$ and $\mathbf{z}$ is mild and hard to interpret). The global $\boldsymbol{\beta}$ is able to control shared features among random groups of samples. The key point in the experiment 4.1 is that if we fix $d$, the global information tuned by $\boldsymbol{\beta}$ in a concrete cluster is clearly interpretable.
>  The motivation to our design is to let a unique $\boldsymbol{\beta}$ control the basic Gaussian mixture priors (means and covariances). But we fully agree that this proposal is interesting. Correlating samples within the same cluster through a different global variable actually generalizes our model, with potential improved cluster interpretability. This model has not been considered so far, but it is certainly something we plan to do in the near future.
> Thank you for the suggestion.
>
> References:
>
> [1] Diane Bouchacourt, Ryota Tomioka, and Sebastian Nowozin. Multi-level variational autoencoder: Learning disentangled representations from grouped observations. In Thirty-Second AAAI Conference on Artificial Intelligence, 2018.
>
> [2] Hosoya, H. (2019, August). Group-based Learning of Disentangled Representations with Generalizability for Novel Contents. In IJCAI (pp. 2506-2513).
>
> [3] Gatys, L. A., Ecker, A. S., & Bethge, M. (2016). Image style transfer using convolutional neural networks. In Proceedings of the IEEE conference on computer vision and pattern recognition (pp. 2414-2423).

---

> > ### Comment · AnonReviewer3 · 2020-11-20
> > **Precision about "local" vs. "global"**
> >
> > Thanks for your explanations.
> >
> > I think however my initial question about beta and the local/global was probably unclear, because you have not answered it, so let me rephrase it.
> >
> > You insist that common features of the datapoint can be extracted in the beta variable, and represented at the global level, rather than the local one. From which, my question is: which of the features of a given dataset can be expected to be found at the global level, rather than the local one?
> >
> > If there is some way to anticipate which feature is learned at the global level (beta) vs the local one (z), what is the mechanism that encourages this separation? And if not, what does the beta variable provide, compared to a model d -> z -> x ?

---

> > > ### Author Response · Authors · 2020-11-22
> > > **Clarifying the mechanism that leads to a separation in local/global latent spaces**
> > >
> > > Thank you for your comment. We apologize if your question was not correctly understood. With the aim at answering the question, we have updated the paper with a new version, including a paragraph in Section 3.3 with an explanation of the mechanism that encourages a separation between local and global spaces, and with the advantages that including the $\boldsymbol{\beta}$ variable provide, with respect to previous approaches. We extend this explanation below:
> > >
> > > The global latent space controls how the $K$ components of the mixture are distributed, as the $K$ networks that obtain $p(\mathbf{Z} | d, \boldsymbol{\beta})$  with $d=1:K$, are fed with the sample of $\boldsymbol{\beta}$. In GMVAE (Figure 1 (b)), each observation comes from an independent sample of $\boldsymbol{\beta}$, and thus, for each observation, the mixture might change. Within each optimization step, the algorithm changes the $\boldsymbol{\beta}$ space by using only local information (from one observation) in order to improve the cluster representation.
> > >
> > > In contrast with GMVAE, in UG-VAE, $\boldsymbol{\beta} $ is shared by a group of observations, therefore the parameters of the mixture are the same for all the samples in a batch. In this manner, within each optimization step, the encoder $q(\boldsymbol{\beta} | \mathbf{X}, \mathbf{Z})$ only learns from the global information obtained from the product of Gaussian contributions of every observation, with the aim at configuring the mixture to improve the representation of each datapoint in the batch, by means of $p(\mathbf{Z} | \mathbf{d}, \boldsymbol{\beta})$ and $p(\mathbf{X} | \mathbf{Z}, \boldsymbol{\beta})$. Hence, the control of the mixture is performed by using global information. In contrast with ML-VAE (whose encoder $q(C_G | \mathbf{X})$ is also global, but the model does not include a mixture), in UG-VAE, the $\boldsymbol{\beta}$ encoder incorporates information about which component each observation belongs to, as the weights of the mixture inferred by $q(\mathbf{d} | \mathbf{Z})$ are used to obtain $q(\boldsymbol{\beta} | \mathbf{X}, \mathbf{Z})$. Thus, while each cluster will represent different local features, moving $\boldsymbol{\beta}$ will affect all the clusters. In other words, modifying $\boldsymbol{\beta}$ will have some effect in each local cluster. As the training progresses, the encoder $q(\boldsymbol{\beta} | \mathbf{X}, \mathbf{Z})$ learns which information emerging from each batch of data allows to move the cluster in a way that the ELBO increases.
> > >
> > > In Figure 3, we have fixed $d$ to obtain samples from the same cluster varying both $\mathbf{z}$ and $\boldsymbol{\beta}$, in order to facilitate the interpretability about which information is captured in both local and global spaces. The results show that $\boldsymbol{\beta}$ is capable of encoding different factors within each cluster. Further, with the aim at extending these results, we have incorporated a similar experiment without fixing $d$ in the supplementary material.
> > >
> > > When training UG-VAE with a single dataset, apart from some intuitions, we do not know a priori which type of information will be encoded in the global space, specially when training such a heterogeneous dataset like CelebA. When training MNIST we can expect that the handwriting style aspects might be encoded by the global space, as for each number, we can change those aspects independently of which number is. The promising result is that we can control these attributes at both local and global levels.
> > >
> > > When training the algorithm with several datasets, as demonstrated in Figures 5 and 6, we expect that the global space will separate clear style aspects that characterise each of them.

---

> > > > ### Comment · AnonReviewer3 · 2020-11-25
> > > > **Still unclear role for beta**
> > > >
> > > > Thanks for your answer. It however does not answers positively my concerns.
> > > >
> > > > My reasoning is as follows: $\beta$ is shared between all elements of the batch, therefore we can only expect it to extract information that is shared between all elements of the batch. In ML-VAE, there is the guarantee that all elements in the batch belong in the same group, allowing the model to extract into $\beta$ information that is common to said group. In contrast, UG-VAE forms batches randomly, so such regularities cannot be present.
> > > >
> > > > From that, I can only formulate a few guesses about what may happen:
> > > >
> > > > 1. either $\beta$ only manages to learn information that is common to the whole dataset. In this case, the variable is probably superfluous, as such information can be encoded in the weight of the neural networks provided they are expressive enough (and it does not seem to be the case given your experiments)
> > > > 2. either by luck, some batches are formed being rather homogeneous from some meta-group. These batches allow the model to extract some information common to multiple datapoints, while the other batches are mostly non-informative as multiple unrelated features are mixed together. So the model still manages to extract something, though at a slower pace.
> > > > 3. either (and this is the most optimistic interpretation) the model uses the information of $d$ to split the latent variable $\beta$ into several subspaces, and each datapoint only contributes to the subspace(s) relevant to its class(es) assigned by $d$. In which case the model learns by itself a construction similar to the structure I described at the end of my original review.
> > > >
> > > > If what happens is the point 3, it should be possible to observe it by analyzing the individual predictions for $q(\beta | \dots)$ of each datapoint, before they are aggregated. If what happens is the point 2, then there is a high risk that the choice of which features end up encoded in $\beta$ and which ends up in $z$ is mostly random, and unreliable from one training to the next.
> > > >
> > > > In my opinion the clarification of this question (what exactly happens at the $\beta$ level, what information is encoded in it, and why) is necessary to properly evaluate the proposed UG-VAE, and so far I don't see it clearly addressed in the submitted article nor in the authors answers.

---

> ### Author Response · Authors · 2020-11-17
> **Response to Reviewer 3: PART 2**
>
> Regarding the small remarks and typos:
>
> -  Corrected the typo on $\Lambda=\Sigma^{-1}$  after Equation (9).
>
> - Corrected the inconsistencies in the KL-Divergence notation.
>
> - We perform a linear interpolation centered on the mean of the generative distributions. The justification for choosing the diagonal on the latent space is for maximizing the variation range across every dimension, and we use $\boldsymbol{\beta}$ values that are in a sphere centered at the prior mean, with radius up to $\sqrt{2\sigma}$ for each dimension. For the global $\boldsymbol{\beta}$, the interpolation goes between $[-1, 1]$ on every dimension, as $p(\boldsymbol{\beta})$ has zero mean. For the local $p(\mathbf{z}  | d, \boldsymbol{\beta})$, as it depends on $\boldsymbol{\beta}$ and $d$, if we denote each dimension of the mean $[\mu_{z0},\ \mu_{z1}, …, \mu_{zd}]$, the local interpolation goes from $[\mu_{z0}-3, \mu_{z1}-3, …, \mu_{zd}-3]$ to $[\mu_{z0}+3, \mu_{z1}+3, …, \mu_{zd}+3]$.
> We have included this justification in the first paragraph of Section 4.1.

---

### Official Review · AnonReviewer4 · 2020-10-28
**Interesting Model but Evaluation Can Be Improved**

**Rating:** 5
**Confidence:** 3

**Review:**

This paper presents a novel deep generative model based on noni.i.d. variational autoencoders that captures global dependencies among observations in a fully unsupervised fashion. The proposed model combines a mixture model in the local or data-dependent space and a global Gaussian latent variable, which captures interpretable disentangled representations with no user-defined regularization in the evidence lower bound. The proposed model is being evaluated in two tasks: (1) disentanglement, and (2) domain alignment.

Pros:
(1) The paper is very well-written and easy to understand. Especially the model figures (figure 1 and 2) are very intuitive and makes it much easier to understand the intuition and difference between the proposed model and previous related works.
(2) The paper is trying to handle a very interesting task, which is to relax the assumption in a lot of previous works in VAE field. The i.i.d data assumption destroys the correlation between data points in the same dataset. Relaxing this constraint will enable wider adoption of this line of models, which can be very useful.


Cons:
My major concern on this paper is the quality of the evaluation section.
(1) first of all, there is no quantitative or qualitative comparison between the proposed method and previous works. Although disentanglement is a task that is hard to quantify, it is hard to show that UG-VAE outperforms other methods.
(2) One of the contribution mentioned is that UG-VAE does not need to tune a bete parameter as in beta-VAEs. But tuning a hyper-parameter for the disentanglement task is not necessarily a negative thing, as UG-VAE also needs to set the total number of lusters. When setting K= 10, it is inducing prior knowledge, and not completely unsupervised anymore.
(3) It is not convincing that the global disentanglement features are learned in the global latent variables. From Figure 3, it just shows some dimensions control certain attributes. It is also not clearly discussed what is defined as "global attributes" and waht is "local attributes".
(4) "Composing graphical models with neural networks for structured representations and fast inference" from NeurIPS 2016 also has a mixture model as latent space. It will provide readers more insight if the authors could discuss the similarities and differences between these two papers.

---

> ### Author Response · Authors · 2020-11-17
> **Response to Reviewer 4**
>
> We thank the reviewer for the careful analysis and positive feedback. We have addressed your suggestions with the aim at improving the evaluations section. Experiments with more datasets and baselines are included. All the answers to your comments are provided below.
>
> (1) Certainly, disentanglement is a task where we lack proper metrics to compare models. With the experiments of 4.1, our goal is to demonstrate the ability of UG-VAE to capture global factors in an unsupervised fashion. Related to the domain alignment experiment, we have expanded the results with two new examples in section 4.2 in which we combine databases of 3D cars, 3D chairs and 2D cars. Note however that in section 4.3 we provide quantitative results, in which we classify using the global projection batches of data points containing different common features.
>
> (2) Unlike beta-VAE, once the generative model is proposed, we do not have to further introduce a hyperparameter in the ELBO lower bound. In UG-VAE, if it were possible, we would like to train our model using maximum likelihood, but that would not be the case in beta-VAE as in that model we intentionally optimize a lower-bound to the ELBO.  We have included this clarification on paragraph below Equation 10 (Section 3.3).
>  We are aware that certain parameters in the generative model (such as the number $K$ of clusters, the $\boldsymbol{\beta}$ dimension and all network parameters) still have to be cross validated, but that also applies to $\beta$-VAE and to any deep generative model.
>
> (3) The concept of local and global, in our work, lies on the fact that the information is extracted from a single image or from a group of images, independently on the semantics of the generative factor. We understand that in other works (like [1], [2], [3]), these concepts are based on the nature of the features, and the global-local features are referred, for example, as style-content. In our experiment, we use the generative model for sampling a batch of images (fixing $d$ to help the interpretations). The reason why we call ‘light’ or ‘hair’ a local factor is that we interpret this disentanglement after varying the local latent variable for every image within a batch, for $d=15$ and $d=7$, respectively. On the other hand, ‘contrast’ is defined as a global factor after interpreting how every image within a batch changes when varying $\boldsymbol{\beta}$. In the second plot top row in Figure 3, we selected the 7th cluster (out of 20) and from samples we can interpret that for $\mathbf{z}$ fixed, the variation of $\boldsymbol{\beta}$ happens to control the presence of beard, hence we assume it is a global factor, and for $\boldsymbol{\beta}$ fixed, the variation of $\mathbf{z}$ controls the amount of hair the person has. This is an interpretation that we draw from samples of the model as we move around each of the two latent spaces.
>
> We have extended the second paragraph of section 4.1 with this explanation.
>
> (4) Thanks for the suggestion. We have included a reference to this paper in Section II. Indeed, in section 3.1, we point out the differences with our work.
>
> References:
>
> [1] Diane Bouchacourt, Ryota Tomioka, and Sebastian Nowozin. Multi-level variational autoencoder: Learning disentangled representations from grouped observations. In *Thirty-Second AAAI Conference on Artificial Intelligence*, 2018.
>
> [2] Hosoya, H. (2019, August). Group-based Learning of Disentangled Representations with Generalizability for Novel Contents. In *IJCAI* (pp. 2506-2513).
>
> [3] Gatys, L. A., Ecker, A. S., & Bethge, M. (2016). Image style transfer using convolutional neural networks. In *Proceedings of the IEEE conference on computer vision and pattern recognition* (pp. 2414-2423).

---

### Official Review · AnonReviewer1 · 2020-10-28
**Experimental section needs more work**

**Rating:** 6
**Confidence:** 4

**Review:**

Response to rebuttal: the authors have drastically improved the quality of the submission with the new experiments and clarifications, I have therefore increased the score to a weak accept.

-------------------------------------------

This paper introduces a non-iid VAE architecture that uses a mixture of gaussian latent space and a global latent variables shared among all the elements of a mini batch to capture global information in correlated datapoints in an unsupervised way.

Overall the paper in well written, and I believe in focuses on two important research directions, namely unsupervised learning of disentangled representations and domain alignment. The model itself is novel and well explained, but I feel the technical explanation is missing intuition on how the model can learn disentanglement in beta from purely random batches, which is not obvious to me.

My biggest concern is in the experimental section, that I did not find convincing enough for a number of reasons:
1. I find it hard to understand if the improvements come from the introduction of the d or the beta latent variables, or a combination of both. How does the model perform in ablation studies in which you remove just one of this components while leaving the others unchanged?
2. In the single-datasets experiment in section 4.1 how do you define what constitutes a local vs global factors?  Currently some of the chosen factors in Figure 3 seem quite arbitrary. Why is light a local factor but contrast a global one? Why is hair local but beard global?
3. The quality of the images is not great to be honest (beta-VAE paper has more convincing ones, just to name a single work), and it is not easy to understand whether the low quality results are due to the fact that as you say you have not validated in depth the networks used or because of flaws in the methodology
4. How would a beta-VAE perform with the same setup of the experiment in 4.1? I would not be surprised if it could capture the same features as your model. It is true as you claim that your method does not require the tuning of the beta hyperparameter in the ELBO, but the UG-VAE needs tuning of the dimensionality of d and beta, and is a more complex architecture than a beta-VAE so it is harder to implement and will take longer to train.
5. It is not clear to me in Figure 4.1. why you are traversing z space in this way, but perhaps I misunderstood what you are doing. How are you guaranteed that you will follow the data manifold? The ML-VAE results might be off just because of this.
6. I believe the more exciting application of this model would be for domain alignment. Why haven't you focused on more multi-datasets experiments?
7. How would a gm-vae baseline with 2 clusters perform with the same setup of the experiment in section 4.2?

In its current state I believe this paper is not ready for acceptance, but I hope the authors will be able to clarify some of my concerns in which case I will increase the score.

Minor comment:
* The second paragraph of the introduction is giving a lot of details on related work. I would recommend to move this discussion to the related work section, and leave the introduction for higher level discussions that only aim at giving intuition to the reader.

---

> ### Author Response · Authors · 2020-11-17
> **Response to Reviewer 1: PART 1**
>
> We thank the reviewer for the detailed feedback and thoughtful recommendation. We have incorporated more detailed explanations and experiments. We address all the concerns below.
> 1. The improvement comes from the combination of both variables. In comparison with ML-VAE (where they feed the model with grouped data), the hidden variable $d$ allows us to infer the group that each sample might belong to when the data is feeded randomly. The latent $d$ can find correlations between samples from different batches by assigning them to the same cluster. If the $d$ is removed, we obtain the ML-VAE, and this type of global information can only be captured in the case we include semi-supervision (Figure 4 corroborates this, as the disentanglement effect of variations in both $\boldsymbol{\beta}$ and $\textbf{z}$ is mild and hard to interpret). The global $\boldsymbol{\beta}$ is able to control shared features among random groups of samples. The key point in the experiment 4.1 is that if we fix $d$, the global information tuned by $\boldsymbol{\beta}$ in a particular cluster is more interpretable. If the global $\boldsymbol{\beta}$ was ignored, we would end up in a standard VAE with Gaussian Mixture priors that is not capable of encoding global information, and thus, neither semi-supervised nor unsupervised approaches could be deployed.
>  We have included a clarification in section 3.1 (second paragraph) with the role that both local $d$ and global $\boldsymbol{\beta}$ play in our model.
>
> 2. The concept of local and global, in our work, lies on the fact that the information is extracted from a single image or from a group of images, independently on the semantics of the generative factor. We understand that in other works (like [1], [2], [3]), these concepts are based on the nature of the features, and the global-local features are referred, for example, as style-content. In our experiment, we use the generative model for sampling a batch of images (fixing $d$ to help the interpretations). The reason why we call ‘light’ or ‘hair’ a local factor is that we interpret this disentanglement after varying the local latent variable and keeping beta fixed for every image within a batch, for $d=15$ and $d=7$, respectively. On the other hand, ‘contrast’ is defined as a global factor after interpreting how every image within a batch changes when varying $\boldsymbol{\beta}$ and keeping fixed local latent variables. In the second plot top row in Figure 3, we selected the 7th cluster (out of 20) and from samples we can interpret that for $\mathbf{z}$ fixed, the variation of $\boldsymbol{\beta}$ happens to control the presence of beard, hence we assume it is a global factor, and for $\boldsymbol{\beta}$ fixed, the variation of $\mathbf{z}$ controls the amount of hair the person has.  This is an interpretation that we draw from samples of the model as we move around each of the two latent spaces.
>  We have extended the second paragraph of section 4.1 with this explanation.
>
> 3. We want to stress the novelty of the design methodology of the probabilistic model, which we show it is able to capture interpretable global effects in an unsupervised way. The fact that this is no longer possible when we disable certain parts of the model to reproduce other proposals in the literature (such as ML-VAE in Figure 4), corroborate our claims.  We believe that this is the main contribution of the paper.
>  Beyond that, certainly a deeper network validation and hyperparameter selection would help to obtain more visually appealing images, but that was not our primary goal, which was always more oriented to latent feature analysis. In this line of work, we could even extrapolate these ideas to other deep generative models such as conditional flow-based models, which have demonstrated superior ability to generate quality images.
>
> References:
>
> [1] Diane Bouchacourt, Ryota Tomioka, and Sebastian Nowozin. Multi-level variational autoencoder: Learning disentangled representations from grouped observations. In *Thirty-Second AAAI Conference on Artificial Intelligence*, 2018.
>
> [2] Hosoya, H. (2019, August). Group-based Learning of Disentangled Representations with Generalizability for Novel Contents. In *IJCAI* (pp. 2506-2513).
>
> [3] Gatys, L. A., Ecker, A. S., & Bethge, M. (2016). Image style transfer using convolutional neural networks. In *Proceedings of the IEEE conference on computer vision and pattern recognition* (pp. 2414-2423).

---

> ### Author Response · Authors · 2020-11-17
> **Response to Reviewer 1: PART 2**
>
> (continuation):
>
> 4. Probably that statement was not redacted on a fair basis, we did not claim that our model is less complex than $\beta$-VAE. However, unlike $\beta$-VAE, once the generative model is proposed, we do not have to further introduce an hyperparameter in the ELBO lower bound. In UG-VAE, if possible, we would like to train our model using maximum likelihood, but that would not be the case in $\beta$-VAE as in that model we intentionally optimize a regularized lower-bound to the ELBO.  We have included this clarification on paragraph below Equation 10 (Section 3.3).
>  We are aware that we still have to cross validate certain parameters in the generative model (such as the number $K$ of clusters, the $\boldsymbol{\beta}$ dimension and all network parameters), but that also applies to $\beta$-VAE and to any deep generative model in general (number of layers, dimension of hidden layers, etc …).
> We would also expect that $\beta$-VAE in 4.1 would capture some disentangled factors common with UG-VAE (actually in the original paper they also use both MNIST and CelebA), but note the additional flexibility of UG-VAE enables to perform domain alignment (4.2) and identifying correlations among data points in the same batch (4.3) with the same generative model. In $\beta$-VAE there is no global space to capture this kind of dependencies.
> We have extended the section “Extended results for section 4.1: Unsupervised Learning of Global Factors” of the Supplementary Material with an analysis of the performance of $\beta$-VAE in a similar setup. As $\beta$-VAE does not include global space, we interpolate only in the latent space $\mathbf{z}$.
>
> 5. We perform a linear interpolation centered on the mean of the generative distributions. The justification for choosing the diagonal on the latent space is for maximizing the variation range across every dimension, and we use latent values that are in a sphere centered at the prior mean, with radius up to $\sqrt{2\sigma}$. For the global $\boldsymbol{\beta}$, the interpolation goes between [-1, 1] on every dimension, as $p(\boldsymbol{\beta})$ has zero mean. For the local $p(\mathbf{z}|d, \boldsymbol{\beta} )$, as it depends on $\boldsymbol{\beta}$ and $d$, if we denote each dimension of the mean $[\mu_{z0}, \mu_{z1}, …, \mu_{zd}]$, the local interpolation goes from $[\mu_{z0}-3, \mu_{z1}-3, …, \mu_{zd}-3]$ to $[\mu_{z0}+3, \mu_{z1}+3, …, \mu_{zd}+3]$. We use the $[-3,3]$ interval as in both CelebA and MNIST the $\mathbf{z}$ variance per cluster that we learn in equation (3) is close to 3.
>  We have included this justification in the first paragraph of Section 4.1.
>
> 6. Our motivation was to illustrate the performance of the model under different settings, but we do also realize that the domain alignment problem is definitely the most exciting application. Following your comment and a similar one by another reviewer, we have expanded these results with two new examples in section 4.2 in which we combine databases of 3D cars, 3D chairs and 2D cars.
>
> 7. In Section “Extended results for section 4.2: Domain Alignment” of the supplementary material, we have included a similar domain alignment experiment, using a GMVAE with two clusters. As GMVAE does not have global variables, the interpolation applies only for the latent encodings in $\mathbf{z}$.  In the latent space, images of different domains are clustered (Image 5 in the supplementary), however, when we interpolate between images from two domains we merely observe a gradual overlap between the two images. Namely, the model is not able to correlate the features of both images, regardless of their domain. On the other hand, with UG-VAE, by keeping fixed the global variable and interpolating in the local one, we maintain the domain but we  translate the features of one image into the other. This analysis corroborates that the model finds this type of correlations in a clearly separated way.
>
> Minor comment:
> - We agree with the reviewer. The paragraph with details on related work have been moved from the Introduction to Section 2.

---

### Author Response · Authors · 2020-11-12
**First revision covering minor changes**

We would like to thank all the reviewers for your feedback and time spent on the paper. During the next few days, we will carefully address each of your comments, covering all your suggestions. In any case, we have updated a first revision of the paper, taking care of your minor remarks and correcting typos.

---

> ### Author Response · Authors · 2020-11-17
> **Major revision**
>
> Dear reviewers,
>
> We have uploaded a new version of the paper addressing all your comments. Experiments with more dataset are provided, and we have extended and clarify some parts according to your suggestions. The supplementary material has also been extended with more experiments. We will include an answer for each reviewer in the corresponding thread. Thank you for your time and consideration.

---

> > ### Author Response · Authors · 2020-11-22
> > **New version, detailing the mechanisms that lead to separation in local/global spaces.**
> >
> > Dear reviewers,
> >
> > We have submitted a new version of the paper that addresses questions from Reviewer 1 and Reviewer 3. In this new version, we add a more detailed explanation in the end of Section 3.3 about the mechanisms that lead to an interpretable separation between the local and global spaces. Further, we have extended the supplementary material with a replication of the experiment 4.1, but without fixing the cluster $d$.

---

### Decision · Program_Chairs · 2021-01-07
**Final Decision**

**Decision:**

Reject

**Comment:**

This paper aims at learning disentangled representation at different level without the supervision signal of group information. To achieve this, the proposed UG-VAE model uses both global variable $\beta$ to represent common information shared across all data, as well as a mixture of Gaussian prior for the local latent variable $p(z) = \int p(z|d)p(d)d$ where $d$ represents the assignment of the group for a particular datapoint. Experiments considered evaluation on unsupervised global factor learning, domain alignment and a downstream application task on batch classification.

Reviewers agreed that the proposed model seems interesting and novel, however some reviewers raised clarity concerns on how to interpret the learned representation by UG-VAE. Revision has addressed this clarity issue to some extent, although some doubts from some reviewers still exists. Also reviewers raised concerns on less competitive experimental results, and the authors have updated the manuscript with improved results.

To me the main issues of the experimental section are (1) no quantitative result is provided regarding global factor learning and domain alignment, and (2) there is no other benchmark being studied in the experimental section. In my view, at least some other VAE representation learning baselines can be included in the batch classification section in order to demonstrate the real benefit of learning global factor based representations in downstream tasks.